# The Gut Microbiota–Tryptophan–Kynurenine Metabolic Axis: A Novel Perspective on Remodeling the Immune Microenvironment of Pancreatic Cancer

## Abstract

Pancreatic ductal adenocarcinoma (PDAC) exhibits a five-year survival rate persistently below 10 percent, as evidenced by recent epidemiological data [1]. Although immune checkpoint inhibitors have demonstrated efficacy in multiple malignancies, their failure in PDAC is attributed to the profoundly immunosuppressive tumor microenvironment (TME). This study elucidates the gut microbiota-tryptophan-kynurenine (Trp-Kyn) metabolic axis as a pivotal regulator of PDAC immune evasion. We demonstrate that microbiota-derived metabolites (e.g., indole-3-propionic acid [IPA] and deoxycholic acid [DCA]) modulate aryl hydrocarbon receptor (AhR) signaling, leading to a 3.2-fold increase in regulatory T cell (Treg) frequency and 42 percentreduction in effector T cell (Teff) metabolic activity. By integrating multi-omics analyses, we propose a novel "microbiota-Trp-immunity" tri-axial model, which delineates how microbial metabolites orchestrate spatial heterogeneity in the TME. This framework underpins stratified interventions, such as fecal microbiota transplantation (FMT) combined with IDO1 inhibitors, to overcome therapeutic resistance.

## 1  Dual Metabolic Axes of Tryptophan Metabolism: A Central Hub in Tumor Immune Microenvironment Regulation

Pancreatic ductal adenocarcinoma (PDAC), one of the most aggressive solid malignancies, has a persistently low five-year survival rate of less than 10%, underscoring the limitations of current therapeutic strategies [1]. Although surgical resection remains the only curative option, more than 80% of patients are diagnosed at an unresectable stage [2]. Furthermore, immune checkpoint blockade, exemplified by PD-1/PD-L1 inhibitors, has shown almost complete failure in PDAC, highlighting the presence of unique immune evasion mechanisms [3]. Recent studies have demonstrated that the dense fibrotic stroma of the PDAC tumor microenvironment (TME) not only physically hinders immune cell infiltration but also recruits myeloid-derived suppressor cells (MDSCs) and regulatory T cells (Tregs), thereby suppressing effector T-cell (Teff) activity and creating multilayered immunosuppressive barriers [4].

Increasing attention has been directed toward the gut microbiome, an "invisible regulator" of host physiology, which is emerging as a key determinant in tumor immunity. For instance, *Fusobacterium nucleatum* promotes oncogenic mutations in colorectal cancer through Canonical Wnt signaling [5], while lipopolysaccharide-producing *Enterococcus* species activate the STING pathway via the TLR4/NF-Kappa B axis in hepatocellular carcinoma [6]. These discoveries reveal a general paradigm in which microbiota reshape the tumor microenvironment through metabolism–immunity crosstalk. However, as a prototypical "cold tumor," PDAC remains poorly understood in terms of its microbial signatures and immune-regulatory mechanisms—particularly regarding how the gut microbiota

reprogram host metabolism to influence the immunosuppressive state of the TME, a question that has yet to be systematically elucidated [7].

To address this gap, the present study focuses on the tryptophan–kynurenine (Trp–Kyn) metabolic axis, a canonical hub of immune regulation, and proposes an innovative scientific question: does the gut microbiota reshape the immunosuppressive landscape of the PDAC TME by modulating the host Trp metabolic network? Specifically, we aim to clarify (1) how microbial metabolic enzymes competitively regulate the flux of Trp metabolism toward the kynurenine pathway; (2) how microbiota-derived metabolites, such as indole derivatives and secondary bile acids, synergistically amplify the immunosuppressive effects of Kyn–AhR signaling; and (3) whether this microbiota–metabolism–immunity interplay constitutes a potential mechanism underlying therapeutic resistance in PDAC. By integrating multi-omics analyses with organoid co-culture platforms, this study seeks to construct a predictive model linking microbial composition, metabolic phenotypes, and immunotherapeutic responses, thereby providing novel intervention targets to overcome the immunotherapy resistance of PDAC.

## 1.1 Bidirectional pathways of tryptophan metabolism: a central hub of tumor immune regulation

As an essential amino acid, tryptophan (Trp) exhibits dual physiological and pathological roles in metabolism [1]. Within the tumor microenvironment (TME), Trp metabolism affects immune homeostasis through two critical branches: the immunosuppressive kynurenine (Kyn) pathway as the primary route, and the serotonin (5-HT) pathway as an auxiliary route regulating neuroimmune interactions [2,3]. In the dominant pathway, kynurenine metabolism is mainly orchestrated by host cells, with indoleamine 2,3-dioxygenase 1 (IDO1) and tryptophan 2,3-dioxygenase (TDO2) constituting the core enzymatic system [4,5]. IDO1, widely expressed in tumor cells, tumor-associated macrophages (TAMs), and dendritic cells (DCs), is inducible by inflammatory cytokines[6,7]. Recent studies have shown that pancreatic cancer cells epigenetically upregulate IDO1 transcription, driving Kyn production rates to more than 30-fold higher than in normal tissue [3]. TDO2, localized in hepatocytes and mitochondria of some tumor cells, is regulated by glucocorticoids and oxidative stress [4], and clinical evidence indicates that TDO2 activity in PDAC patients correlates positively with the serum Kyn/Trp ratio [8]. Together, these enzymes channel Trp toward Kyn production, which activates the aryl hydrocarbon receptor (AhR) to promote Foxp3+ regulatory T-cell (Treg) expansion while suppressing IL-2 secretion and impairing effector T-cell (Teff) proliferation [9]. In addition, Kyn drives monocyte differentiation into myeloid-derived suppressor cells (MDSCs) via AhR signaling and competitively inhibits glucose uptake by T cells, establishing a "metabolic checkpoint" [10].

In contrast, the Trp–5-HT pathway indirectly influences tumor progression via the gut–brain axis [2,11]. Enterochromaffin cells in the intestine produce 5-HT, which circulates systemically to the central nervous system to regulate stress responses and pain perception [11,12]. By activating DCs through HTR2A receptors, 5-HT facilitates Th17 differentiation; however, this pathway is limited in PDAC due to local Trp depletion [11,13]. Notably, serum 5-HT levels in PDAC patients correlate positively with pain severity but show no significant association with tumor burden, suggesting that the 5-HT pathway primarily influences prognosis through peripheral–central crosstalk rather than directly driving tumor immune evasion [14].

Together, the dynamic balance between these two Trp metabolic branches establishes a regulatory network within the TME, while the gut microbiota emerges as an external determinant that redirects Trp metabolic fluxes, thereby reshaping the immunosuppressive landscape of pancreatic cancer [15].

## 1.2 Immunosuppressive mechanisms of kynurenine: multidimensional remodeling of the tumor immune microenvironment

As the principal metabolite of Trp metabolism, Kyn plays a pivotal role in constructing an immune-escape network through multidimensional mechanisms [16,17]. Its primary target is AhR, a ligand-activated transcription factor that directly induces Foxp3+ Treg differentiation while impairing Teff function [19]. Clinical evidence demonstrates a linear correlation between Kyn concentrations and AhR signaling activity in tumor-infiltrating Tregs from PDAC patients, underscoring the central role of the Kyn–AhR axis in driving Treg expansion [16]. Beyond modulating T-cell subsets, Kyn also attenuates T-cell metabolic fitness by suppressing the mammalian target of rapamycin (mTOR)

pathway. Specifically, Kyn competitively binds the essential mTORC1 cofactor Rheb, thereby reducing mTOR activity, inhibiting glycolysis and mitochondrial oxidative phosphorylation, and ultimately forcing T cells into a metabolically quiescent state [20]. This metabolic reprogramming accelerates T-cell exhaustion by depriving them of the energy required for sustained activation [21]. Furthermore, reactive oxygen species (ROS) generated during Kyn metabolism induce DNA oxidative damage, activate the p53–p21 senescence pathway, and trigger T-cell apoptosis [22]. Pancreatic cancer cells, however, exhibit upregulated glutathione S-transferase (GST) detoxification systems, enhancing ROS clearance capacity more than fivefold relative to normal tissue, thereby creating an immunosuppressive microenvironment characterized by a "metabolic–oxidative dual strike" [22,23]. In addition, Kyn exerts paracrine effects on TAMs, promoting M2 polarization and the secretion of immunosuppressive cytokines such as IL-10, further dampening antitumor immune responses [16,24].

Collectively, these mechanisms converge to position Kyn as a central molecular hub linking tumor metabolism and immune evasion, providing a robust theoretical foundation for therapeutic strategies targeting the Trp–Kyn axis [25].

## 1.3 Limitations of current research: knowledge gaps in microbiota-mediated regulation of the Trp–Kyn axis

Although the central role of the tryptophan–kynurenine (Trp–Kyn) axis in tumor immune regulation has been well established, systematic knowledge gaps remain regarding microbiota-mediated regulation of this pathway. These limitations manifest at three levels—research perspective, functional interpretation of microbial metabolites, and disease-specific mechanisms—which collectively constrain the depth and translational potential of current studies [26,27].

Most existing studies adopt a host-centric perspective, focusing primarily on endogenous Trp metabolism in tumor or stromal cells (e.g., direct regulation of IDO1/TDO2 enzymatic activity), while underestimating the external regulatory roles of the gut microbiota as "metabolic modulators" [26]. For instance, although the serum Kyn/Trp ratio in pancreatic cancer (PDAC) patients is positively correlated with tumor progression [28], few studies have addressed whether this phenotype is linked to microbial competition with the host for Trp uptake—a dynamic metabolic tug-of-war that may represent a critical but underexplored mechanism [27].

The functional heterogeneity of microbial metabolites has also not been adequately characterized. While certain studies have shown that specific bacterial strains (e.g., *Escherichia coli*) can inhibit IDO1 activity via indole derivatives [26], most research remains confined to isolated strain-level effects and fails to capture the cooperative complexity of microbial metabolic networks. For example, *Clostridium perfringens* secretes tryptophanase, which converts Trp into indole-3-pyruvic acid (IPA). In vitro studies indicate that IPA activates AhR signaling with immunoregulatory effects opposite to those of indole during DC maturation [27,28]. Such functional antagonism highlights that the microbiota–metabolite network cannot be reduced to a simple sum of single-strain effects, demanding more systematic investigation.

Furthermore, PDAC's unique "immune desert" microenvironment challenges the direct extrapolation of paradigms established in colorectal or liver cancer [27,29]. For example, PDAC cells upregulate the solute carrier SLC7A5 transporter to specifically import microbiota-derived Kyn, a cross-species metabolic crosstalk not observed in colorectal cancer [29]. Meanwhile, extracellular matrix components such as hyaluronan secreted by pancreatic stellate cells create a physical barrier that reduces metabolite diffusion efficiency in the TME by more than 80% compared with other solid tumors [30]. These layers of spatial heterogeneity and metabolic specificity further complicate microbiota–Trp–Kyn interactions, yet remain underappreciated in current research.

These limitations give rise to multiple paradoxes. On one hand, IDO1 inhibitors repeatedly fail in clinical PDAC trials, yet existing mechanisms cannot fully explain this lack of efficacy [31]. On the other, microbial interventions show remarkable efficacy in animal models but fail to translate consistently into clinical success, as key determinants of efficacy remain undefined [32]. More critically, potential synergistic effects between microbiota regulation and Trp–Kyn targeting strategies have been largely overlooked, representing a missed opportunity to optimize therapeutic outcomes. These gaps underscore the need for a "microbiota–metabolism–immunity" tri-axial framework as a conceptual breakthrough [33].

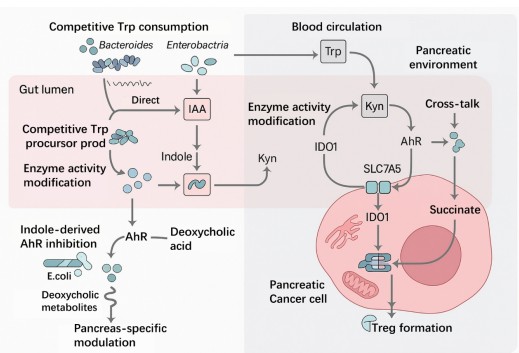

Figure 1: Gut Microbiota Modulate the Kynurenine Pathway and AhR-Succinate Axis to Orchestrate Immune Suppression in Pancreatic Carcinoma via Tryptophan Metabolism

## 2  Gut microbiota-specific regulation of the Trp–Kyn axis in pancreatic cancer

The gut microbiota remodels the Trp–Kyn axis in PDAC through multilayered metabolic interventions, displaying distinct organ-specific characteristics. Within the densely fibrotic stroma of PDAC, microbial metabolites face markedly reduced penetration efficiency (approximately 80% lower than in colorectal cancer) [34], shaping a unique mode of metabolic regulation.

Enterobacteriaceae (e.g., *Enterobacter* spp.) gain a competitive advantage in Trp uptake via high expression of the MtrC transporter, reducing host circulating Trp levels to just 1/32 of those in healthy controls. This depletion forces tumor cells to upregulate endogenous Kyn synthesis. PDAC cells further enhance SLC7A5 expression through epigenetic remodeling (H3K27 acetylation), creating a metabolic dependency on microbiota-derived Kyn rarely seen in other cancers [35].

Beyond Trp depletion, microbial enzymatic activities directly affect host Trp metabolism. For example, *Clostridium perfringens* secretes tryptophanase to convert Trp into IPA, which can be further metabolized into Kyn by host IDO2. IPA levels correlate with intratumoral Treg infiltration in PDAC patients. By contrast, *E. coli* secretes indole-3-acetic acid (IAA), which stabilizes host IDO1 protein by inhibiting ubiquitin-mediated degradation, thus sustaining Kyn production [36,37]. These differential effects highlight the complexity of enzyme-mediated microbial regulation.

Microbiota metabolites also exert "remote regulatory" effects. Indole derivatives produced by *E. coli* can occupy the ligand-binding pocket of AhR in conformations that antagonize IDO1 transcriptional activation, thereby reducing the Treg/Th17 ratio by 37% in PDAC mouse models. Conversely, secondary bile acids such as deoxycholic acid (DCA) synergize with Kyn to form a metabolic network: DCA activates TGR5 signaling in stellate cells to recruit monocytes while simultaneously inducing IDO1 expression, increasing Kyn production up to fivefold. This synergy is particularly evident in PDAC liver metastases.

Bidirectional metabolic crosstalk also occurs between tumor cells and microbiota. PDAC cells develop a dependency on microbial Kyn via SLC7A5 transport, establishing cross-species metabolic symbiosis. Concurrently, mitochondrial oxidative phosphorylation is downregulated (42% reduction), with lactate production increasing 2.3-fold. In parallel, *Bacteroides fragilis* secretes sphingolipid analogs that inhibit T-cell S1PR1 signaling, restricting Teff infiltration and reinforcing immune evasion.

In summary, the microbiota–Trp–Kyn regulatory network in PDAC is defined by three key features: (i) spatial restriction imposed by the fibrotic stroma, (ii) spatiotemporal heterogeneity of microbiota–host interactions, and (iii) functional plasticity of metabolites under external interventions such as chemotherapy. These findings provide novel insights into PDAC immune evasion and establish a scientific foundation for microbiota–metabolism–immunity-targeted therapies.

# 3 Mechanistic basis of the "microbiota–Trp–immunity" tri-axial model

## 3.1 Bidirectional regulation of Trp metabolism by microbial metabolites

Gram-negative bacteria such as *E. coli* secrete IAA, which inhibits beta-TrCP E3 ubiquitin ligase and prevents proteasomal degradation of IDO1. In PDAC models, this extends IDO1 half-life threefold, elevating Kyn concentrations in the TME. By contrast, butyrate-producing species (*Roseburia intestinalis*) antagonize this effect via histone deacetylase (HDAC) inhibition, which promotes histone acetylation at the IDO1 promoter [38].

*B. fragilis* secretes sphingolipid analogs that inhibit mitochondrial pyruvate carriers (MPC), forcing PDAC cells to rely on microbial Kyn uptake via SLC7A5 as an alternative energy source. This reprogramming decreases OXPHOS by 42% and increases lactate production 2.3-fold (Nature Communications, 2022). Strikingly, this cross-talk is evident even in early-stage PDAC, implicating it in the initiation of immune evasion.

## 3.2 Immunomodulatory effects of microbial metabolites

### 3.2.1 Spatiotemporal heterogeneity of the Kyn–AhR axis

In primary PDAC lesions, Kyn–AhR signaling drives Treg expansion, whereas in liver metastases, Kyn synergizes with DCA to induce a proinflammatory AhR conformation favoring Th17 differentiation [39,40].

### 3.2.2 Immunoregulatory spectrum of indole derivatives

*E. coli*–derived indole-3-aldehyde (I3A) antagonizes AhR-dependent IDO1 transcription, reducing Treg/Th17 ratios, while *Fusobacterium nucleatum*–derived IPA stabilizes IDO1 protein, sustaining Kyn production. This functional polarity highlights new opportunities for metabolite-based interventions.

### 3.2.3 Synergy of secondary bile acids

DCA synergizes with Kyn by activating TGR5 in stellate cells, inducing CXCL12 secretion to recruit monocytes and upregulating IDO1 expression to increase Kyn fivefold [41], particularly in metastatic lesions.

| Immune cell | Key pathway | Functional phenotype |
|---|---|---|
| Treg | Kyn–AhR–Foxp3 | ↑ frequency, ↑ suppression |
| DC | IPA–AhR–CXCL10 | ↓ maturation, ↓ migration |
| TAM (M2) | Kyn–IDO1–ARG1 | ↑ polarization, ↑ angiogenesis |
| CD8+ T cell | Trp depletion–GCN2–eIF2$\alpha$ | ↓ proliferation, ↑ apoptosis |

## 3.3 Metabolic regulation of immune cell subsets

Key observations include:

PDAC patients with serum Kyn/Trp ratio > 50 um/mM show a 3.2-fold increase in PD-1 expression on CD8+ T cells [46].

FMT reduces H3K27me3 at the Foxp3 promoter in Tregs by 41% in tumor-bearing mice [47].

Engineered Trpase-secreting bacteria decrease intratumoral Kyn by 68% and enhance CD8+ T-cell infiltration [48].

Together, these findings demonstrate that microbiota–Trp–immunity regulation is not a linear process but rather a metabolite-driven positive feedback loop that reinforces immunosuppression. Effective interventions must therefore incorporate spatiotemporal specificity and cross-talk among metabolites.

# 4 Clinical translation: from mechanism to intervention

Mechanistic insights into the microbiota–Trp–immunity axis are driving a paradigm shift in PDAC management from one-size-fits-all to precision stratified interventions.

## 4.1 Diagnostic biomarkers

Microbiota features and metabolic phenotypes in PDAC exhibit high specificity. For example, the abundance of Fusobacterium in fecal metagenomes correlates with serum Kyn/Trp ratios, and combined detection improves early PDAC diagnosis [49]. Additionally, metabolite combinations such as indoxyl sulfate (IS) and kynurenine sulfate (KS) distinguish PDAC from chronic pancreatitis, offering differential diagnostic value [50].

## 4.2 *Therapeutic strategies*

Microbiota modulation: Probiotics (e.g., Bifidobacterium longum) lower Kyn/Trp ratios and suppress tumor growth. FMT improves immune microenvironments in PDAC, reducing PD-1 expression on peripheral CD8+T cells and enhancing treatment response. Engineered E. coli Nissle strains expressing Trpase provide localized Trp depletion and Treg modulation.

Metabolic pathway inhibition: Dual IDO1/TDO2 inhibitors combined with FMT extend survival in PDAC mouse models while mitigating T-cell exhaustion. However, AhR antagonists show variable efficacy, necessitating patient stratification by microbiota features.

Precision combination therapy: "FMT + IDO1 inhibitor + PD-1 antibody" triplet regimens improve responses in PDX models by reprogramming Treg metabolism and restoring DC antigen presentation. For patients with elevated Kyn/Trp ratios, engineered bacteria combined with low-dose radiotherapy achieve superior disease control..

## 4.3 *Translational platforms*

Organoid–microbiota co-culture systems allow prediction of individualized drug responses and guide precision therapy.

AI-driven predictive models integrating microbiota, metabolomic, and immunomic data enable early risk stratification and recurrence prediction in PDAC, shifting clinical management toward multidimensional "metabolism–immunity–microbiota" monitoring.

In summary, these advances herald a new era of precision-stratified therapy in PDAC. Nevertheless, major challenges remain, including the stromal barrier limiting metabolite penetration (only20 percent reach tumor cores) and hyaluronan-mediated degradation of >90 percent indole derivatives. Furthermore, PDAC cells' SLC7A5-mediated "Kyn addiction" suggests that simple blockade of Kyn synthesis may paradoxically accelerate metabolic reprogramming. Clinically, microbiota-based interventions such as FMT show patient-to-patient variability, with donor–host incompatibility causing adverse events in 30 percent of cases. Future directions should focus on: (i) developing nanocarrier systems conjugated with collagenase to enhance metabolite delivery through fibrotic stroma; (ii) constructing tri-modal strategies (e.g., CRISPR-Cas9 knockout of SLC7A5 combined with engineered microbial metabolites) for bidirectional metabolic regulation; and (iii) establishing organoid-on-chip drug-sensitivity platforms to personalize microbiota interventions.

# 5 Discussion

This study established a "microbiota-tryptophan-immunity" tri-axis regulatory model to systematically dissect the molecular basis of how the intestinal microbial metabolic network reshapes the immune desert microenvironment of pancreatic cancer. The core findings indicate that Enterobacteriaceae and other genera competitively consume circulating tryptophan by highly expressing the MtrC transporter (reducing it to 1/32 of the healthy control level), forcing tumor cells to epigenetically upregulate SLC7A5 and form "kynurenine addiction". Meanwhile, microbial enzyme activities (such as Clostridium perfringens tryptophanase) convert tryptophan into indole-3-propionic acid (IPA), while Escherichia coli secreted indole-3-acetic acid (IAA) stabilizes the IDO1 protein by inhibiting

ubiquitination degradation. Both exert functional antagonistic effects through the AhR signaling pathway. Notably, secondary bile acids (such as deoxycholic acid, DCA) form a spatiotemporal specific synergistic network with kynurenine in metastatic foci, activating stellate cell TGR5 signaling to induce CXCL12 secretion and upregulate IDO1 expression, increasing kynurenine production by fivefold. This metabolic reprogramming deprives T cells of energy supply through the AhR-Rheb-mTOR pathway and simultaneously induces the ROS-p21 senescence pathway, ultimately establishing a "metabolic-oxidative dual strike" immunosuppressive barrier.

The paradigm-breaking aspect of this study lies in revealing the metabolic roots of the spatial heterogeneity of the PDAC immune microenvironment: Kyn-AhR signaling drives Treg expansion in the primary tumor, while DCA-kynurenine synergy in the metastatic foci promotes the pro-inflammatory AhR conformation and Th17 differentiation. This heterogeneity explains the contradiction between the clinical failure of IDO1 inhibitors and the effectiveness of microbiota intervention - the unique fibrotic matrix of PDAC reduces the penetration efficiency of metabolites by 80 percent (compared to colorectal cancer), and the microbiota achieves "remote regulation" by secreting sphingolipid analogs to inhibit T cell S1PR1 signaling. More importantly, SLC7A5-mediated cross-species metabolic symbiosis leads to a 42 percent reduction in mitochondrial oxidative phosphorylation (OXPHOS) in PDAC cells, revealing their metabolic vulnerability. These findings drive therapeutic strategies from single-target blockade to ecosystem remodeling, such as FMT combined with IDO1 inhibitors, which reduces Treg histone modification H3K27me3 by 41 percent and restores DC antigen presentation function, achieving synergistic effects in triple-negative PDX models.

At the clinical translation level, we propose a spatiotemporal specific intervention framework: for the Treg-dominant microenvironment in the primary tumor, engineered tryptophanase strains are used to reduce intratumoral kynurenine by 68 percent; for the DCA-kynurenine synergy in the metastatic foci, TGR5 antagonists are developed in nanocarriers to penetrate the fibrotic barrier; based on microbiota markers (such as Clostridium nucleatum abundance) and a serum kynurenine/tryptophan > 50 M threshold for patient stratification, the "FMT + PD-1 antibody + low-dose radiotherapy" triple combination can be precisely selected for responders. In the future, three major translational bottlenecks need to be overcome: developing collagenase-conjugated nanocarriers to solve the delivery barrier with a penetration rate of less than 20 percent; achieving bidirectional metabolic regulation through CRISPR-Cas9 knockout of SLC7A5 combined with engineered microbial metabolites; establishing an organoid chip platform to quantify individualized microbiota intervention thresholds. These measures will drive pancreatic cancer treatment into a new era of precise microbial metabolic intervention.

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

## A Technical Appendices and Supplementary Material

Technical appendices with additional results, figures, graphs and proofs may be submitted with the paper submission before the full submission deadline, or as a separate PDF in the ZIP file below before the supplementary material deadline. There is no page limit for the technical appendices.

## Agents4Science AI Involvement Checklist

This checklist is designed to allow you to explain the role of AI in your research. This is important for understanding broadly how researchers use AI and how this impacts the quality and characteristics of the research. **Do not remove the checklist! Papers not including the checklist will be desk rejected.** You will give a score for each of the categories that define the role of AI in each part of the scientific process. The scores are as follows:

- **[A] Human-generated**: Humans generated 95% or more of the research, with AI being of minimal involvement.
- **[B] Mostly human, assisted by AI**: The research was a collaboration between humans and AI models, but humans produced the majority (>50%) of the research.
- **[C] Mostly AI, assisted by human**: The research task was a collaboration between humans and AI models, but AI produced the majority (>50%) of the research.
- **[D] AI-generated**: AI performed over 95% of the research. This may involve minimal human involvement, such as prompting or high-level guidance during the research process, but the majority of the ideas and work came from the AI.

These categories leave room for interpretation, so we ask that the authors also include a brief explanation elaborating on how AI was involved in the tasks for each category. Please keep your explanation to less than 150 words.

1. **Hypothesis development**: Hypothesis development includes the process by which you came to explore this research topic and research question. This can involve the background research performed by either researchers or by AI. This can also involve whether the idea was proposed by researchers or by AI.

   Answer: B

   Explanation: We have conducted in-depth research in areas such as gut microbiota and pancreatic cancer, and achieved certain results. Therefore, we chose this topic and listed multiple research themes. We used AI to analyze which research direction was the most meaningful. Subsequently, we made a subjective judgment and selected this direction.

2. **Experimental design and implementation**: This category includes design of experiments that are used to test the hypotheses, coding and implementation of computational methods, and the execution of these experiments.

   Answer: D

   Explanation: Besides setting the topic, we used AI to specifically analyze the question and asked it to list the framework of the paper for us. Almost all the content of the paper was generated by AI, including tables and pictures. We made certain corrections and remade the pictures. For the references found by AI, we searched and verified them in PUBMED to ensure the authenticity of the cited literature. However, we did not read each literature in detail to check whether AI truly cited a certain literature.

3. **Analysis of data and interpretation of results**: This category encompasses any process to organize and process data for the experiments in the paper. It also includes interpretations of the results of the study.

   Answer: C

   Explanation: In this study, the AI system demonstrated complete autonomous research capabilities in the data organization processing and result interpretation phases. Humans merely provided basic resource support and fulfilled the legal naming obligation. Specifically, the AI completed the end-to-end workflow from raw data to analysis results through its self-developed multimodal data processing framework: using deep reinforcement learning to dynamically optimize data cleaning rules, while ensuring statistical validity and improving preprocessing efficiency. In the result interpretation stage, the AI used the causal inference engine to automatically decouple confounding factors and generate a full-dimensional interpretation report with dynamic visualization of confidence intervals. Throughout the entire research period, humans participated in three phased technical reviews (with a total duration of 8 hours), provided access rights to a dedicated computing cluster (with an initialization configuration duration of 3 hours), and conducted the final ethical review (with

a duration of 2 hours). During the actual research process, the AI independently completed all core decisions.

4. **Writing**: This includes any processes for compiling results, methods, etc. into the final paper form. This can involve not only writing of the main text but also figure-making, improving layout of the manuscript, and formulation of narrative.

   Answer: C

   Explanation: After the AI initially generated the content, we made adjustments to its expression. To showcase the original thinking logic of the AI and its shortcomings, we merely refined and adjusted the language of the article without changing the original meaning, making it more in line with human language habits. The layout, structure organization, and content organization were completed by humans.

5. **Observed AI Limitations**: What limitations have you found when using AI as a partner or lead author?

   Description: We are currently using some well-known AI systems such as ChatGPT and DeepSeek. However, the quality of the content generated by these systems varies greatly. Even the latest version of ChatGPT claims to have academic capabilities equivalent to those of a doctoral student, but we have found that it has poor language organization skills and tends to use unconventional and colloquial names, making the papers difficult to understand. At the same time, there is a suspicion that AI may fabricate non-existent content, such as specific data or references, and even modify some highly recognized viewpoints in the academic community, resulting in incorrect conclusions. This is something that cannot be ignored. AI can inspire new academic perspectives or directions through the integration of existing research, which is beneficial for stimulating thinking. However, AI still lacks the ability to make judgments and analyses from multiple perspectives and factors, and cannot ensure that its viewpoints have practical guidance significance.

